# A Novel Technique for Autograft Preparation Using Patient-Specific Instrumentation (PSI) Assistance in Total Hip Arthroplasty in Developmental Dysplasia of Hip (DDH)

**DOI:** 10.3390/jpm13091331

**Published:** 2023-08-29

**Authors:** Chun-Ru Lin, Hsuan Chou, Chu-An Luo, Shu-Hao Chang

**Affiliations:** 1Department of Medical Education, Chang Gung Memorial Hospital, Linkou Branch, No. 5, Fuxing Street, Guishan District, Taoyuan City 333423, Taiwan; asd450132@gmail.com; 2School of Medicine, College of Medicine, Fu Jen Catholic University, No. 510, Zhongzheng Road, Xinzhuang District, New Taipei City 24205, Taiwan; eric158273@gmail.com; 3Department of Precision Surgery Development, A Plus Biotechnology Co., Ltd., 6F, No. 23, Qiaohe Road, Zhonghe District, New Taipei City 23529, Taiwan; robers.luo@aplusbio.com; 4Department of Orthopedics, Fu Jen Catholic University Hospital, Fu Jen Catholic University, No. 69, Guizi Road, Taishan District, New Taipei City 24352, Taiwan

**Keywords:** patient-specific instrumentation, developmental dysplasia of hip, total hip arthroplasty, peri-operative outcome, improved complication rate

## Abstract

Due to the change in the structure of the proximal femur and acetabulum in patients with developmental dysplasia of the hip, total hip arthroplasty (THA) was difficult to perform for surgeons. To elevate the acetabular coverage rate, we developed a technique in the use of a patient-specific instrumentation (PSI) graft in patients with developmental dysplasia of hip (DDH) undergoing surgery. This study aims to evaluate the peri-operative outcomes of THA with PSI graft in patients with DDH. This study recruited 6 patients suffering from Crowe I DDH with secondary Grade IV osteoarthritis. All the patients underwent THA with PSI graft performed by a well-experienced surgeon. Perioperative outcomes included surgical procedures, blood loss during operation, the volume of blood transfusion, length of hospitalization, complications, and the mean difference in hemoglobin levels before and after surgery. All the outcomes analyzed were assessed by mean and standard deviation. The average duration of the surgical procedure was found to be 221.17 min, with an SD of 19.65 min. The mean blood loss during the operation was 733.33 mL, with an SD of 355.90 mL. The mean length of hospital stay was calculated to be 6 days, with an SD of 0.89 days. Furthermore, the mean difference between the pre- and postoperative hemoglobin levels was 2.15, with an SD of 0.99. A total of three patients received 2 units of leukocyte-poor red blood cells (LPR) as an accepted blood transfusion. There were no reported complications observed during the admission and one month after the operation. This study reported the peri-operative outcomes in the patients with DDH who underwent THA with PSI graft. We found that THA with PSI graft would provide a safe procedure without significant complications. We assumed that the PSI graft in THA may increase the coverage rate of the acetabulum, which may increase the graft union rates. Further cohort studies and randomized controlled trials were needed to confirm our findings.

## 1. Introduction

Developmental dysplasia of the hip (DDH) describes a disorder of normal hip development resulting in neonatal instability, acetabular dysplasia, hip subluxation, and dislocation of the hip [1]. The frequency of the new child patient in one year is 1–7% in the UK, Canada, and north-eastern Germany from the period 1995–2012, which indicates DDH is one of the most common childhood disabilities [2].

In the management of DDH, osteoarthritis secondary to dysplasia is the indication for total hip arthroplasty(THA), which can improve Harris Hip scores and pain [3]. However, acetabular cups loosen because the malposition of the prosthesis remains in many patients [4]. Thus, patient-specific instrumentation (PSI) was newly developed and attracts surgeons’ interest in the accuracy of the procedure [5]. PSI was intended to guide the physical positioning of the prosthesis in THA [6]. PSI was first introduced for total knee arthroplasty (TKA) and shoulder arthroplasty. In total knee arthroplasty, PSI improves the accuracy of the femoral component and global mechanical alignment but might increase the risk of outliers for the tibial component alignment, while it can deliver better outcomes of operative time and blood loss in TKA [7,8]. THA using PSI assistance was also developed for better accuracy of hip surgery than conventional THA and the postoperative functional outcomes [9,10,11,12].

The procedure for THA with PSI requires 3D preoperative planning using a computed tomography (CT) scan or magnetic resonance imaging (MRI) to forecast the size of the cup and the stem, which was proven accurate and can shorten the learning curve of operators using the minimally invasive direct anterior approach [13]. The 3D computer model was built with three-dimensional reconstruction software. The optimal acetabular inclination and anteversion were at the discretion of the operators before the operation. The point of reference and target position or orientation of the implants could be customized to fit the patients’ and operators’ requirements [14]. The guide and acetabular model were then 3D-printed and installed in the patients according to the plan. The laser was used to assist in the installation of the guide and the placement of the acetabular component. The CT scan was performed to follow up on the outcomes [14,15].

In this case series study, we aimed to understand the perioperative outcomes of DDH patients undertaking THA with PSI graft in Asia, including operation time, the day of hospital stay, blood loss, hemoglobin variation, and complication rates.

## 2. Materials and Methods

### 2.1. General Data

This case series study was approved by the institutional review board of Fu-Jen Catholic University Hospital (FJUH112270). The data in this study was collected from January 2018 to January 2023 and involved 6 female cases who underwent THA with PSI procedures, including 3 cases for both left and right developmental dysplasia of the hip (DDH). The period of each study case includes the time of the operation to the last follow-up. The selection criteria for PSI procedures were revised by a high-volume surgeon with over 150 THA cases per year. All patients underwent imaging examinations, including X-rays and CT scans. Patients in this study suffered from Crowe I DDH with secondary grade IV osteoarthritis and had minimal response to previous conservative treatment, and PSI-assisted THA was suggested. Conservative treatments include pain management, rehabilitation, and supplement consumption.

### 2.2. PSI Graft Preparation

Based on individual CT images, the 3D models of the pelvis and femur were reconstructed and demonstrated in Figure 1. The femoral head and neck were resected under simulation and repositioned at the acetabulum according to the surgeon’s guide. Once the position of the resected femoral head was determined, the osteotomy could be designed for the best fit between the contact surfaces of the acetabular and osteotomized femoral head. The patient-specific cutting jig was designed to fit the patient’s proximal femur with six pin holes for stable fixation and was composed of the guiding slots that were generated in line with the osteotomy. The jig was then manufactured by a commercial 3D printer with medical-grade nylon. Before clinical use, moist heat sterilization was implemented to ensure that no visible deformation happens on the cutting jig.

### 2.3. Treatment Procedures

The patients were positioned in the decubitus position with the operated side positioned upwards, and a standard posterior approach was made to access the diseased hip. The femoral head and neck were dislocated initially, following which a 3D-printed PSI (patient-specific instrument) cutting jig was applied to the posterior aspect of the femoral head and neck. The cutting jig was secured to the femur using three to four 2 mm Kirschner wires that were inserted into the pin holes on the jig. The femoral head was then cut using a reciprocating saw along the pre-planned grooves on the cutting jigs, thereby obtaining the pre-planned bone graft. Subsequently, the bone graft was placed on the dome of the acetabulum, and the contact surface was further fine-tuned using a rongeur. The bone graft was then removed, and the acetabulum was reamed to the desired size. The bone graft was temporarily secured to the acetabular dome using two Kirschner wires, and the bone graft was further reamed using a reamer. Finally, the desired cup was applied to the acetabulum and fixed using acetabular screws. The bone graft was ultimately secured using 2–3.5 mm compression screws, with or without washers. All the surgical procedures were conducted by a highly experienced surgeon.

### 2.4. Postoperative Treatment

The patients were advised to limit weight-bearing activities for 6 weeks following the surgical procedure. Full squatting training was initiated 6 weeks after the operation.

### 2.5. Outcome Assessment

The peri-operative outcomes were evaluated based on several parameters, including the duration of the surgical procedure, blood loss during operation, volume of blood transfusion, length of hospitalization, complications, and the mean difference in hemoglobin levels before and after surgery. The calculation of the hemoglobin difference was based on the measurements taken on the day before and following the surgical intervention. All patients were required to attend follow-up appointments at the outpatient department during the week following the operation. Any complications that arose during this time were documented for analysis. All outcomes were analyzed using the Microsoft™ Excel™ 365 MSO 16.0.13528.203018 64 bit software developed by Microsoft Corporation, based in Redmond, Washington, USA, and presented in terms of mean and standard deviation (SD).

## 3. Results

Table 1 presents the patients’ characteristics, including patient information, chief complaints, diagnosis, Crowe’s classification, and underlying medical conditions. The patients were all female, and only one of them is overweight (BMI > 25). The pain was the primary complaint reported by all six cases, with four cases reporting weakness, two cases reporting the limited range of motion (ROM), and two cases reporting limping gait. According to the standard classification system, all six cases were classified as Crowe’s type I. Underlying medical conditions were identified in three cases, including anemia due to leiomyoma, left leg poliomyelitis, and eczema.

Before the total hip arthroplasty, the surgical plan was simulated for every patient, and the patient-specific instrument was printed for the surgery in Figure 2. Preoperative and postoperative X-ray images of the hip were collected and are included in Figure 3. Table 2 presents the peri-operative outcomes, including the mean and SD of several parameters. The average duration of the surgical procedure was found to be 221.17 min, with an SD of 19.65 min. The mean blood loss during the operation was 733.33 mL, with an SD of 355.90 mL. The mean length of hospital stay was calculated to be 6 days, with an SD of 0.89 days. Furthermore, the mean difference between the pre-and postoperative hemoglobin levels was 2.15, with an SD of 0.99. A total of three patients received 2 units of leukocyte-poor red blood cells (LPR) as an accepted blood transfusion. Notably, there were no reported complications observed after the operation. No implant loosening, stem sinking, graft loosening, or dislocation was found during the follow-up period.

## 4. Discussion

This study evaluated the peri-operative outcomes of patients with DDH and advanced OA who underwent THA with PSI graft. The peri-operative outcomes comprised the duration of the surgical procedure, blood loss during the operation, volume of blood transfusion, length of hospitalization, complications, and mean difference in hemoglobin levels before and after the surgical intervention. The demographic features of DDH were also collected. The demographic data of patients undergoing total hip arthroplasty using patient-specific instrumentation (PSI) include sex, age, height, body weight, BMI, primary complaint, diagnosis, Crowe’s classification, and comorbidity.

This study collected adult patients from the age of 35 to 69 years old, and they were all Asian. The implications of DDH could last from children to adults if no diagnosis or intervention were made. Several studies demonstrate the epidemiology of DDH. The incidence of DDH varies from an average of 0.006% in Africans in Africa in the period 1966–1977, 0.076% in Japan in 2011–2013, 4.45% in Poland in 2013–2018, and 7.6% in Native Americans in 1950–1982 due to ethnicity, race, age of the population, diagnostic criteria, and screening protocol (physical examination, X-ray, or ultrasound) [16,17,18].

The patients suited for this study are all females, which is one of the risk factors for DDH. The known risk factors also include firstborn, breech position, family history of hip abnormalities, oligohydramnios, macrosomia, limited hip abduction, talipes, and swaddling [18]. DDH appears more common in females with a female-to-male ratio of 6:1 [19]. The reason for that was the increased ligamentous laxity appearing shortly as a result of the circulating maternal hormone relaxin. The anatomic location of DDH was found more in the left hip (60%) than in the bilateral hip (20%) because the left hip is more likely to adducted against the mother’s lumbosacral spine in the intrauterine position (left occiput anterior), where less cartilage covers the bone of the acetabulum [20].

Symptoms and signs of the patients in this study were pain, weakness, and limited range of motion, which were the top few manifestations of DDH. DDH could be first detected by history taking and physical examination, and the different manifestations of DDH vary in different ages. In patients less than 3 months old, DDH could present palpable hip subluxation/dislocation on Barlow, Ortolani, and Galeazzi (Allis) tests and hip click sounds. In 3-month-old to 1-year-old patients, limitations in hip abduction, lower limb length discrepancy, and dislocations on the Klisic test could be observed. In patients who are more than 1 year old, DDH could present pelvic obliquity, lumbar lordosis, Trendelenburg gait, and toe-walking. In adults with DDH, the increased internal rotation with the hip in flexion may appear, and provocative tests may show anterior apprehension signs and positive on prone external rotation tests [21,22,23,24]. The most common presentation of symptomatic hip dysplasia is most localized to the groin (72%) with insidious onset and secondary to the lateral aspect of the hip (66%). Lateral-based pain and a limp appear due to abductor fatigue and prolonged activity [25].

Current diagnostic methods include ultrasounds, AP-view radiographs, arthrograms, and Computed Tomography (CT). All of the patients in this study were diagnosed via X-ray or CT, and Crowe’s classification for each patient was determined [1].

In general, conservative and surgical management were used for DDH patients. Non-surgical management included weight loss, movement modification, Non-Steroidal Anti-Inflammatory Drugs (NSAIDs), physical therapy, and intra-articular corticosteroid injections [26]. Surgical management may include hip arthroscopy adjunct to periacetabular osteotomy, salvage pelvic osteotomy, hip resurfacing, and total hip arthroplasty [27,28,29]. Complications of DDH surgeries include sciatic nerve palsies, nonunion with greater trochanter osteotomy, increasing risks of hip dislocation after arthroplasty, hip subluxations, periprosthetic femur fracture, and infection [30,31]. Avascular necrosis of the femoral head could be observed after closed reduction for DDH due to an excessive abduction of the hip [1,32]. Due to advanced osteoarthritic change secondary to DDH, the patients in this study received pilot surgical treatment, namely, PSI-assisted total hip arthroplasty, which may increase the precision of the bone graft preparation and decrease complications such as bone graft nonunion in theory.

The surgical approach to DDH in adults was rather difficult due to multiple factors, especially the malpositioning of the cup [33]. The overall pooled incidence of dislocation of primary THA was 2.10% from a systemic review collecting 149 articles with data on 4,633,935 primary THAs and 35,264 dislocations from 1975 to 2019 [34]. The risk factors for post-THA dislocation also include over 70-year-olds, low income, white ethnicity (only when compared with Asian ethnicity), drug use disorder, and body mass index (BMI) > 30 kg/m^2^ [35]. Medical factors, including neurological disorders, psychiatric disease, comorbidity indices, and previous surgery may increase the risk of dislocation [36]. Higher Crowe grade, intra-operative fracture, and post-operative acetabular offset less than 16 mm in previous surgeries could also be the risk factors of post-THA dislocation [37]. Protective factors of dislocation include surgical factors such as the anterolateral, direct anterior, or lateral approach, and the posterior approach with short external rotator and capsule repair; implant factors, such as larger femoral head diameters, elevated acetabular liners, dual mobility cups, cemented fixations, and standard femoral neck lengths; and hospital-related factors such as experienced surgeons and high surgeon procedure volume [34]. The Lewinnek safe zone of 15° ± 10° of anteversion and 40° ± 10° of inclination has been considered the goal of THA [38]. The use of the PSI was reported to reduce the risk of malpositioning of the hip outside of the safe zone, and the peri-operative outcomes may also improve [15].

PSI-assisted THA was used to treat the patients with Crowe I DDH and secondary grade IV osteoarthritis in this study, and it could be one of the pilot studies for a new option of management. There were no dislocation or other complications in patients with Crowe I receiving PSI-assisted THA in this study. In the patients with DDH Crowe II to IV, previous randomized controlled trials show significant improvement in THA with PSI compared to the conventional surgical methods. A previous clinical trial with 20 hip surgeries from 17 patients with Crowe II/III DDH undergoing THA with PSI reported that there were no significant differences in anteversion and inclination between pre- and postoperative plans. A significant improvement in the mean score of HHS was noted at 12 weeks and 24 weeks postoperatively compared to the preoperative mean scores [39]. Another clinical trial, with 104 patients with DDH Crowe I to IV, which compared the peri-operative outcomes between PSI and conventional operation groups, reported that all postoperative indexes of cup orientation and positioning in patients with DDH Crowe III and IV were significantly improved, which demonstrated the benefits of using PSI as a better approach to DDH. However, the outcomes of PSI in patients with DDH Crowe I was not yet determined. There was no significant difference in the accuracy of PSI for patients with DDH Crowe I [40]. The indication of accuracy included cup placement and orientation. The peri-operative outcomes of patients should also be considered in the outcomes of the PSI. Our studies would provide data for further evaluations of the peri-operative outcomes in patients with DDH Crowe I undertaking THA with PSI.

Blood loss and volume of blood transfusion are two of the indicators in peri-operative outcomes of patients who underwent PSI [15]. Blood loss in our patients was within 1250 mL, and the standard protocol of 2 unit blood transfusion was only used for our patients with blood loss above 850 mL. There are several disadvantages to increased blood loss and transfusion. First of all, increasing blood loss and transfusion could lead to extra costs, about USD 343.63 per unit of RBCs for patients in the US [41]. Secondly, blood transfusion can also lead to transmission of an infectious process, transfusion-related acute lung injury (TRALI) [42], or other complicated complications such as acidosis, hypothermia, and coagulopathy, which are associated with a high mortality rate [43].

Hemoglobin levels before and after the surgical intervention should also be monitored for the prediction of transfusion. Hb levels 8.0 and 8.9 g per dL might trigger the transfusion for both first allogeneic transfusions and postoperative transfusions [44]. The post-operative Hemoglobin levels in the patients of this study are all above 8.9, which demonstrates blood loss without triggering post-operative transfusion.

The duration of the surgical procedure and the length of hospitalization are also two of the peri-operative outcomes [15]. Previous studies reported that long operation time increased the risk of infections [45] and had a higher possibility of discharge to nursing health services or rehabilitation facilities instead of going home [46]. Long operation time may be associated with increased length of hospitalization [47]. The duration of operation time in THA with PSI is still controversial and might differ for patients with different Crowe classifications. In the study of simulation THA with PSI in 2020, the PSI group did not prolong the duration of the surgical procedure [48]. In another study in 2021, the operation time in patients with DDH Crowe III and IV is less in the PSI group than conventional surgery group, but the operation time in patients with DDH Crowe I was longer in the PSI group than that in the conventional group. The reasons might be that the anatomical structure in patients with DDH Crowe I is relatively simple, and extra procedures are needed when performing PSI on patients with DDH Crowe I [40]. However, with the successful experiences of PSI-assisted THA on patients with DDH Crowe I in this study, there is a higher chance to achieve shorter operation time for DDH Crowe I patients and to reduce related risks.

The PSI in DDH for THA could increase the accuracy of placing the cup orientation and lower the threshold for junior physicians. In total, 48 junior physicians were included in a randomly assigned THA simulation on the DDH hip model with or without PSI. The inclination differences between the PSI and non-PSI groups were significantly smaller in cup inclination and anteversion. The final cup size in the PSI group completely meets the planned cup size [48]. This result demonstrates the better accuracy of PSI in THA.

A previous study on the subjective feelings of patients undergoing THA with or without PSI demonstrated the benefits of PSI. The subjective opinions toward THA with PSI of patients with DDH were better than that without PSI. The Harris Hip Score, Oxford University Hip Score (OHS), Visual Analogue Scale (VAS) score, patient satisfaction score, amount of bleeding, and postoperative complications incidence were analyzed, and there were no significant differences between the PSI group and the traditional THA group, which described the equivalent benefit outcome of these two groups. The Forgotten Joint Score (FJS-12) in the PSI group was significantly better than that in the control group [49].

There were some limitations in this study. First, the study demonstrated the successful THA with PSI graft from patients in Crowe’s type I DDH. However, it was difficult to identify early-stage DHH with the need for THA, because the symptoms of early-stage DDH in young adults are rather mild or almost obscured [50]. Secondly, there were six patients with DDH in this study. Due to the limited number of participants, the outcomes of the patients may not apply to the others, such as the higher Crowe stage. The long-term benefits of PSI to the patients with Crowe I DDH were yet determined, but the preliminary outcomes were satisfactory in this study and promising for future studies. Confounding bias should be excluded for the causation of the benefits of PSI. Therefore, these patients should be monitored in the long-term period, or randomized control trials should be performed. The Harris Hip score and the analysis of the images should be included. Further cohort studies and randomized controlled trials are needed to confirm our findings.

## 5. Conclusions

This study reported the peri-operative outcomes including the duration of the surgical procedure, blood loss during operation, the volume of blood transfusion, length of hospitalization, complications, and the mean difference in hemoglobin levels before and after surgery in the patients suffering Crowe I DDH with secondary advanced OA under THA with PSI graft. We found THA with PSI graft was safe with a low complication rate. Further long-term follow-up, cohort studies, and randomized controlled trials are needed to confirm our findings. A study of THA with PSI graft in higher Crowe-grade patients would be needed to expand the usage.

## Figures and Tables

**Figure 1 jpm-13-01331-f001:**
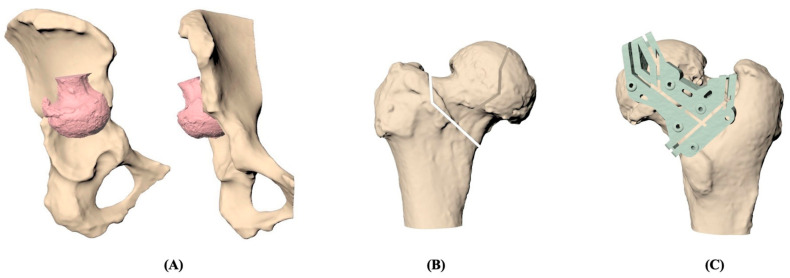
**The 3D model reconstruction process of total hip arthroplasty using patient-specific instrumentation (PSI).** (**A**) Preoperative reconstruction of pelvis and femur. (**B**) Determination of the position of osteotomy in the femoral head. (**C**) Simulation of the 3D-printed patient-specific cutting jig on the femoral head.

**Figure 2 jpm-13-01331-f002:**
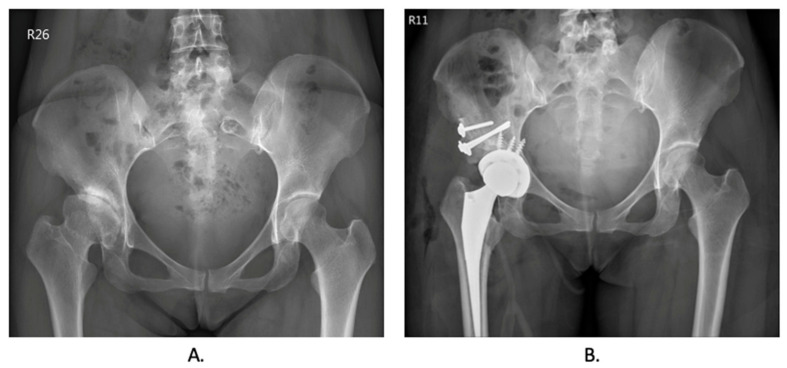
**Representative radiographic image of preoperative and postoperative hip undergoing total hip arthroplasty.** (**A**). Preoperative X-ray image of developmental dysplasia of the hip. (**B**). Postoperative X-ray image while undergoing total hip arthroplasty using patient-specific instrumentation.

**Figure 3 jpm-13-01331-f003:**
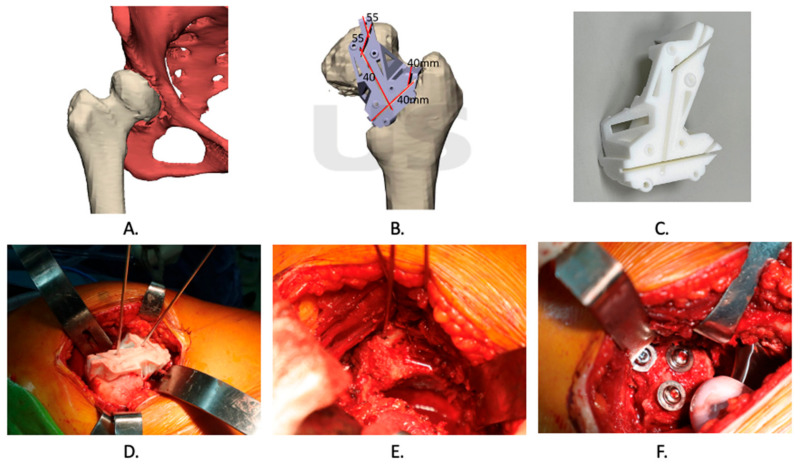
**Surgical plan and procedure.** (**A**). The 3D reconstruction of the hip region. (**B**). Simulation of patient-specific jig and cutting line. (**C**). The 3D-printed patient-specific jigs. (**D**–**F**). Implementation of total hip arthroplasty using the patient-specific jig.

**Table 1 jpm-13-01331-t001:** Demographic data of patients undergoing total hip arthroplasty using patient-specific instrumentation (PSI).

No.	Sex	Age (Years Old)	Height (cm)	Body Weight (Kg)	BMI	Primary Complaint	Diagnosis	Crowe’s Classification	Comorbidity
1	F	44	161	56	21.6	Pain, weakness	Left DDH * with advanced OA	Crowe I	Leiomyoma, leiomyoma-related anemia
2	F	59	147.8	59	27	Pain, weakness	Right DDH * with advanced OA	Crowe I	Left leg poliomyelitis
3	F	43	160	53	20.7	Pain, weakness, limited ROM	Left DDH * with advanced OA	Crowe I	Eczema
4	F	40	160	62	24.2	Pain, limping gait	Left DDH * with advanced OA	Crowe I	None
5	F	35	166	59	21.4	Pain, weakness, limited ROM	Right DDH * with advanced OA	Crowe I	None
6	F	69	154	52	21.9	Pain, limping,	Left DDH * with advanced OA	Crowe I	None

* DDH means developmental dysplasia of the hip.

**Table 2 jpm-13-01331-t002:** Perioperative outcomes of patients undergoing total hip arthroplasty using patient-specific instrumentation (PSI).

No.	Sex	Operation Time (min)	Blood Loss (mL)	Anesthesia	Length of Hospitalization (Day)	Blood Transfusion (U)	Pre- Operation Hb (g/dL)	Post- Operation Hb (g/dL)	MD * (g/dL)	Complication
1	F	220	1250	General anesthesia	5	2U	9.6	8.9	0.7	No complication
2		190	350		7	X	15.1	11.8	3.3	
3		225	450		5	X	13.1	10.5	2.6	
4		237	500		6	X	13.5	10.6	2.9	
5		245	850		7	2U	12.4	11.1	1.3	
6		210	1000		6	2U	13.6	11.5	2.1	

* Mean difference in Hb before and after surgery.

## Data Availability

The authors declare that the data supporting the findings of this study are available within the paper.

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
