# Peer review of "A Novel Technique for Autograft Preparation Using Patient-Specific Instrumentation (PSI) Assistance in Total Hip Arthroplasty in Developmental Dysplasia of Hip (DDH)"

_jpm, 2023, doi:10.3390/jpm13091331_

Round 1

Reviewer 1 Report

The paper, in general, is well written, but the Introduction Section is too longer. In my opinion the first paragraph should be shortened, because it explain dysplasia in general, and the work is focus in patients with osteoarthritis secondary to hip displasya. 

Although the work is interesting, I think that the Patient-Specific-Instrumentation should be applied for complex cases, I mean Crowe III-IV, because for Crowe I, the anatomy is quite similar to the normal anatomy.

Another problem is the short serie. It is only 6 patientes.

I woyuld like that the authors use the system for more complex cases. In this scenario, 6 patiens is valid.

Author Response

Point 1:

The first paragraph should be shortened, because it explains dysplasia in general, and the work is focused on patients with osteoarthritis secondary to hip dysplasia.

Response 1:

Thank you for your valuable feedback. For the introduction, a precise explanation of dysplasia has been well-written in the new version of the manuscript and the focus was put on the osteoarthritis secondary to hip dysplasia. We had revised the manuscript. (line 41-45)

Developmental dysplasia of the hip (DDH) describes a disorder of normal hip development resulting in neonatal instability, acetabular dysplasia, hip subluxation, and dislocation of the hip.[1] The frequency of the new child patient in one year is 1-7% in the UK, Canada, and North-Eastern Germany from 1995-2012, which indicates DDH is one of the most common childhood disabilities.[2]

Point 2:

Although the work is interesting, I think that the Patient-Specific-Instrumentation should be applied for complex cases, I mean Crowe III-IV, because for Crowe I, the anatomy is quite similar to the normal anatomy. Another problem is the short series. It is only 6 patients. I would like the authors to use the system for more complex cases. In this scenario, 6 patients are valid.

Response 2:

The primary objective of this article is to present the preliminary results of our innovative idea, which explores the feasibility of utilizing 3D-printed PSI cutting jigs in the preparation of femoral head bone grafts. Due to the exploratory nature of our study, the number of cases included was limited. After establishing the viability for Crowe I deformities, we have expanded our efforts to address more complex deformities, such as Crowe III-IV. As we accumulate more cases involving intricate deformities, we intend to submit further findings and share our insights with the academic community.

Reviewer 2 Report

The manuscript needs several improvements. For example,

- lines 93 to 101 should be removed.

- in table 1, mention units for age, height, body weight.

- mention figure 1 in the main text.

- "the study was conducted from January 2018 to January 2023". The presented results on post-operative complications may reflect on follow-up for one or more patients.

Quality of English needs to be improved - like lines 160 and 161 "The patients' characteristics was demonstrated in Table 1. The patients were female, and only one of them are overweight".

Author Response

Response 1:

Thank you so much for your valuable opinion.  We’ve addressed the related issues mentioned above and corrected them accordingly. We mentioned the post-operative complications in the manuscript. (line 151-152)

 No implant loosening, stem sinking, graft loosening or dislocation was found during the follow-up period.

Point 2:

Quality of English needs to be improved - like lines 160 and 161 "The patients' characteristics was demonstrated in Table 1. The patients were female, and only one of them are overweight".

Response 2:

Thank you so much for your suggestion.  We’ve corrected them and related sentences.

Round 2

Reviewer 1 Report

The modifications have been done correctly.

Reviewer 2 Report

The manuscript version-2 is much improved compared to the initial version..